# Early Notice Pointer, an IoT-like Platform for Point-of-Care Feet and Body Balance Screening

**DOI:** 10.3390/mi13050682

**Published:** 2022-04-27

**Authors:** Florina Silvia Iliescu, Ling Tim Hong, Jin Ming Jaden Toh, Mirela Petruta Suchea, Octavian Narcis Ionescu, Ciprian Iliescu

**Affiliations:** 1School of Applied Science, Republic Polytechnic, 9 Woodlands Avenue 9, Singapore 738964, Singapore; tohjinming@gmail.com; 2National Institute for Research and Development in Microtechnologies (IMT-Bucharest), 077190 Bucharest, Romania; mirasuchea@hmu.gr (M.P.S.); octavian.ionescu@imt.ro (O.N.I.); 3School of Engineering, Republic Polytechnic, 9 Woodlands Avenue 9, Singapore 738964, Singapore; hong_ling_tim@rp.edu.sg; 4Department of Electrical and Computer Engineering, Center of Materials Technology and Photonics, Hellenic Mediterranean University, 71410 Heraklion, Greece; 5Faculty of Electrical and Mechanical Engineering, Petroleum and Gas University of Ploiesti, 100680 Ploiesti, Romania; 6Academy of Romanian Scientists, 010071 Bucharest, Romania; 7Faculty of Applied Chemistry and Biotechnologies, University “Politehnica” of Bucharest, 011061 Bucharest, Romania

**Keywords:** screening posture, foot arch, POCT, prophylaxis, IoT, personalised medicine

## Abstract

Improper foot biomechanics associated with uneven bodyweight distribution contribute to impaired balance and fall risks. There is a need to complete the panel of commercially available devices for the self-measurement of BMI, fat, muscle, bone, weight, and hydration with one that measures weight-shifting at home as a pre-specialist assessment system. This paper reports the development of the Early Notice Pointer (ENP), a user-friendly screening device based on weighing scale technology. The ENP is designed to be used at home to provide a graphic indication and customised and evidence-based foot and posture triage. The device electronically detects and maps the bodyweight and distinct load distributions on the main areas of the feet: forefoot and rearfoot. The developed platform also presents features that assess the user’s balance, and the results are displayed as a simple numerical report and map. The technology supports data display on mobile phones and accommodates multiple measurements for monitoring. Therefore, the evaluation could be done at non-specialist and professional levels. The system has been tested to validate its accuracy, precision, and consistency. A parallel study to describe the frequency of arch types and metatarsal pressure in young adults (1034 healthy subjects) was conducted to explain the importance of self-monitoring at home for better prevention of foot arch- and posture-related conditions. The results showed the potential of the newly created platform as a screening device ready to be wirelessly connected with mobile phones and the internet for remote and personalised identification and monitoring of foot- and body balance-related conditions. The real-time interpretation of the reported physiological parameters opens new avenues toward IoT-like on-body monitoring of human physiological signals through easy-to-use devices on flexible substrates for specific versatility.

## 1. Introduction

Approximately 1.71 billion people are diagnosed with musculoskeletal disorders (MSD), limiting mobility and dexterity, causing disability, and leading to a low quality of life worldwide. The highest prevalence of lower back pain among MSD patients, 568 million people, is the leading contributor to MSD-related disability in 160 countries, raising concerns mainly because the prevalence increases with age, although young people are also affected, and the incidence increases more rapidly in low-income and middle-income countries. Moreover, it is generally acknowledged that pronated (over-pronated, low arch) or supinated (high arch) feet are among the leading causes of pain and injury in the foot and ankle area [1] and, over time, in the rest of the body [2,3]. The improper biomechanics [4,5] are progressive and alters the gait, load on foot [6], walking speed, body posture, and balance [7]. Overexertion injuries to the musculoskeletal system, static postures, repetitive movements at work, and pain are reported as a financial burden [8,9].

Regrettably, people of all ages present for clinical assessment only when irreversible changes and chronic conditions have occurred [10] when diagnosis procedures are sophisticated and the therapeutic approaches are tardive, complicated, or only palliative. Generally, clinical assessment methods of body balance include static, quasi-mobility, and mobility tasks. The Berg Balance Scale (BBS), Dynamic Gait Index (DGI) [11,12], Clinical Test of Sensory Integration and Balance (CTSIB) [13], Functional Obstacle Course (FOC) [14], Performance-Oriented Mobility Assessment—gait (M-POMA) [15], Postural Stress Test (PST) [16], Solid Ground, Balance Board, Rotating Platform, Horizontal Translational Platform, Treadmill, and Computerised Dynamic Posturography are complementary and depend on the specific technology. The Computerised Dynamic Posturography Sensory Organisation Test (CDP-SOT), the gold standard [17], supported the validation of postural control measurements [18] in healthy adults [19,20] and patients with multiple sclerosis [21], stroke [22], transtibial amputation [23], and low back pain [24].

Specialised literature acknowledged that other non-standard balance assessment methods have their equivalent numerical assessment parameters, and there is little similarity between these parameters and between different methods of corresponding evaluation [17]. Some platforms equipped with force transducers recorded the ground reaction forces, a centre of pressure (COP) and a centre of gravity (COG), on assessed subjects while they stood on them. In foot-ankle biomechanics, strain gauges are often used to measure the 

Strain of joints, muscles, ligaments, and plantar soft tissues [25]. Clinicians may use scored balance performance of body sway on a platform for patients [26,27] and healthy individuals from various age groups [28,29] to diagnose postural stability and evaluate the related risks. However, using these devices successfully depends on patients’ compliance because the testing, interpretation, and follow-ups require repeated trips to specialists’ clinics that are not always readily available. Furthermore, it is difficult to compare the results between the studies using various methods due to the lack of standardisation. For instance, examining working posture could be inconsistent because of the nature of the tasks performed which influences the decision about the assessment method. In the case of observation-based examinations, the postural assessment includes the observation of static images or single video frames for diagnosis and preventive decisions. There are in place methods for musculoskeletal disease assessment accompanied by posture categories that partition posture ranges for the trunk, shoulder, elbow, forearm, and wrist [30]. Consequently, remote and virtual posture analysis have emerged as proactive ergonomic assessment programs to consolidate the clinical applications of the microtechnological processes [31]. One ergonomic assessment based on Internet of Things (IoT) wearable devices was designed to identify, analyse, prevent, and control risk factors for manual material handling. The device acquires real-life data that translates into measures to maintain a quality posture [32]. Meanwhile, the Industrial Internet of Things was used to assess ergonomic indexes in near-real-time and avoid classical procedures which involve time-consuming analysis [32,33]. IoT has also been applied for remote medical monitoring [34]. A supervised learning approach was built to acquire, process, and store data for posture analysis during sleep studies [35]. Asymmetric sitting biomechanics has been overviewed with the help of a seat cover that employed novel pressure sensing architecture as an endpoint application. Using this device to monitor sitting posture can help correct posture and prevent health problems while transforming the usually expensive and time-consuming method into a 30 min IoT-based procedure [36]. IoT-type devices that comprise thin sensors are the next step in point of care testing (POCT) and personalised medicine [37] for tailored diagnostics and therapy that addresses each patient’s needs based on their specific background, disease prognosis, and assessed risks [38]. Despite complex and multifactorial difficulties [39], considerable progress has been made: the Federal Drug Administration (FDA) has approved drugs labelled on specific genomics biomarkers [40], tests for the long-term genetic-based management of breast cancer [41], emerging titration schemes tested for specific medication [42], and dispensing devices designed for diabetic or Parkinson’s Disease patient support [43]. Nowadays, e-skin sensors are designed and developed to measure and display physiological variables such as heart rate, blood oxygen saturation, glucose, or moisture. Moreover, the capabilities of transparent [44] or semitransparent [45] layer-based devices extends to most of the sensing organs of the human body [46] to detect colourless and odourless gasses [47] and vibration-, respiration-, sound-, and pulse changes [48]. The analysis of biomarkers and stimuli occurs in a network of e-skin sensors. For instance, a flexible sensor tag can noninvasively monitor surface temperature for precise diagnostics and feedback treatment in the longterm [49,50,51,52]. They are cheaper than commercially available inertial Measurement Unit (IMU)-based systems as well as non-invasive and highly stretchable which makes them more comfortable for long-term use and thus, a suitable sensing technology for developing continuous, out-of-hospital, real-time monitoring and management systems for lower back pain [53,54]. Frediani et al. described a system consisting of two dielectric elastomer (DE) sensors arranged on shoulder straps and custom-made wireless electronics designed to measure the capacitance of the sensors and calibrate them when the user wears them for the first time [55]. However, the evolving measurement technology relevant for posture-related diagnostics and prevention is based on environmental, cultural, and economic backgrounds [56]. The preference for existing formats can affect the compatibility of the vast technology available for examination methods and be costly and time-consuming. Merging the existing features of various technologies and methods is the criterion of a modern approach. A cost-effective and simple-to-use scanner at home would allow remote electronic device-based assistance for real-time evaluation, continuous monitoring, and altering of therapeutic approaches. Personal electronic communication devices connected to a scanner and supported by software-based technology would allow rapid measurements for timely postural analysis. Furthermore, users of various professional and cultural backgrounds could also follow simple instructions to conduct the assessment with or without face-to-face supervision if they do not need special training.

We, therefore, report the development of a wireless platform—Early Notice Pointer (ENP) for the self-assessment of bodyweight, body load distribution on both the right and left forefoot and rearfoot, and early detection of body imbalance. The designed and developed system can measure and map the weight shifting of users as pre-assessment data. The prototype comprises pressure sensors coupled with specific electronic elements to detect and transmit data to a mobile phone and, based on a predetermined setpoint, to display results graphically and numerically for straightforward interpretation and distant communication with the specialists. The prototype has been internally validated against available marketed devices for clinical compliance. The ENP determines user weight distribution and can alert the user when the weight distribution is above or below average level. Therefore, it assists its users with an informed decision regarding their presentation for specialist consultation and specific treatment. Furthermore, the ENP is designed for everyone who wishes to monitor their bodyweight, posture, and plantar pressures: healthy individuals who may detect any affected biomechanics before any clinical signs and symptoms begin and patients who wish to monitor and detect any complications of their biomechanics-related conditions in a timely manner. Since it offers wireless data transmission and emits an early warning regarding the bodyweight distribution cum balance, it could work as an IoT-like device for point of care testing (POCT) for personalised medicine. In conclusion, the ENP is a potential screening device for the home-based evaluation of bodyweight, balance, and plantar pressure for effective prophylaxis of biomechanics-related conditions.

## 2. Materials and Methods

Methodology refers to the design of the prototype and the testing for validation. Additionally, a study of foot arch and balance discusses variables such as bodyweight, body height, and carrying bags concerning foot arch and posture.

### 2.1. The Design of the Prototype

The ENP resembles normal weighing scale functions complemented by the capability to compare the load distributions on a person’s feet and calculate the relative imbalance of the load distributions. Figure 1 describes the working principle and the essential components of the ENP. The user steps on the platform comprised of load cells to measure the force applied corresponding to the distributed weight. The sensors’ placement allows readings that define eight regions to describe the entire foot. The load was observed to set up the average profile for the prototype (green colour in Figure 1) against MatScan. The system uses Wi-Fi technology to send signals to a backend database for further analysis and accommodate multiple measurements. The display on the user’s mobile phone and computer provides a practical and valuable platform for easy analysis and transmission of the results. Moreover, it can issue an early warning if readings are outside the pre-established thresholds and help the user correct their posture and bodyweight or call for specialist advice.

#### 2.1.1. Materials Used to Design the Prototype

Force Sensor Plates: Two identical force plates with compression load cells (ISVASIA SINGAPORE PTE LTD), used for measuring the left and right foot weight loads.Signal Processing: PCB with 8 INA125P (a 16-pin Instrumentation Amplifier) (Texas Instruments) to filter, amplify, and remove noise.Arduino Mega Development board with Wireless transmitter: uses 8 out of 16 analogue inputs (Arduino LLC).Cellphone/tablet Display: User interface for the measurement, analysis, and display of results.Remote Raspberry Pi for MySQL Databased: A pocket PC for hosting Database services for storage records for post-analysis (Adafruit Industry).

#### 2.1.2. The Procedure to Design the Prototype

Figure 2 shows the assembly of the electronic and mechanical elements.

The screening platform is designed as a flat plate that displays the measurement results in a simple and direct manner. The eight sensory cells collect point weight readings and are distributed symmetrically over the four corners of each foot to collect pressure data from corresponding areas of the right and left foot. The circuits and sensory cells were joined for stability since the platform is transportable. The ENP has a specific workflow (Appendix A) for users’ guidance, correct data acquisition, and analysis to ensure consistent and accurate results. Monitoring the procedure will be under the direct supervision of users or helpers, and the results will be available via mobile phone to both the users and the specialists either locally or at the remote clinics. The proposed sensitive and easy-to-use screening platform has been prototyped (Figure 2) and tested for consistent data acquisition. A mobile application (App) and a website display were developed to present the testing results. Figure 3 illustrates the mobile App-based and website-based displays. The colour code display is advanced and allows the results recorded by the device to be easily interpreted by the user and specialists.

The software can be used in website and mobile phone modes. The website mode starts once the sensing platform and the computer are connected. Premeasurement setup checks are performed automatically to ensure the sync with the database. Furthermore, manual inputs via Home Tab are allowed to correct the initiation. Once the initiation is performed, the Home page displays the twelve most recent user data, as weight chart, percentage by the part chart, weight table, and percentage table (Figure 3a,b). Data is also displayed on Data Chart and Table pages as charts and tables with an option to Load more data for both Charts for weight and percentage by location to explain the load distribution for the forefoot and rearfoot on both the right and left sides and the overall values.

The Mobile phone mode starts upon installing the App, pairing with the Bluetooth, and initialising the check to display the weight and percentage distributions. The platform was programmed using a prototype development board that provided connectivity and a battery shield and enabled easy troubleshooting, fast prototyping, and proof-of-concept.

### 2.2. The Testing of the Prototype

The prototype evaluation followed internal and external validation which involved human subjects. To be eligible to participate in the validation study, participants had to be healthy with no known foot biomechanics-related problems. The study was approved by the Republic Polytechnic Singapore Institutional Review Board (IRB), written informed consent was obtained from all participants, and IRB guidelines were followed in all procedures with human participants (IRB approval for MOE2013-TIF-2-G-036; Assigned HSR Code: SAS-F-2016-002).

#### 2.2.1. Demographics

The cutoff was set to determine the level of distribution based on the data collected from 150 subjects (normal distribution in a healthy population, the sample size is sufficient for the statistical significance of the pilot study). The group studied was a population aged 19 to 70 years with no orthopaedical or podiatric medical history, including no injuries and no deviation of the foot arch.

#### 2.2.2. Data Acquisition

The design ensures that demographic and anthropometric parameters do not influence the results. First, the ENP has a specific workflow (Appendix A) and uses physiological parameters such as age- and gender-standardised bodyweight and foot arch charts to ensure the consistency of the interpretation of collected data. Second, testing the collected bodyweight readings at 55 kg, 58 kg, 70 kg, and 83 kg against the voltage changes was performed. The linear relationship (R^2^ = 1) showed the consistency of the signal and confirmed the final placement of sensors and circuit processing units. From the users’ perspective, monitoring the procedure will be under the direct supervision of users or helpers, and the results will be available via mobile phone to both the users and specialists.

Two stages of testing and validation (as a proof-of-concept study) were conducted to compare the prototype’s performance with commercial low to high-end equipment for internal and external validation. The newly created prototype has been tested for validation against specific equipment: two commercial, personal weighing scales (max 150 kg) and two foot scanners (MatScan from Tekscan and RSscan pressure plate foot scanners from RSscan International).

The first stage was the internal validation to compare the weight output given by the prototype with the one given by validated weight scales. The first stage of validation on multiple commercial weighing scales was performed to calculate and compare the standard deviation of the weight output from the prototype. A user’s bodyweight was measured 50 times. Further tests were conducted to reduce the variance in weight with multiple types of scale. Eventually, the testing compared the prototype with the most stable scale.

In the second stage, testing for external validation was conducted to validate the weight distribution accuracy against two commercially available professional feet scanning equipment. Therefore, the prototype was used to measure plantar pressure and body sway. Three measurements were performed for each of the enrolled subjects upon informed consent. The foot arches were observed to set up the average profile and the setpoint for the prototype (green colour in Figure 1) against MatScan. The data from 30 tests on both prototype and commercial equipment for foot and balance scans were used.

#### 2.2.3. Data Analysis

Data were collated and analysed for significance using a *t*-test (the measure of a difference between the scanners compared) to validate our product’s performance against the two validated foot scanners (Matscan and RS FootScan). The significant preliminary variance was corrected with further fine-tuning to stabilise the device. The measurements were compared with the developed prototype’s results and modifications were implemented to improve the prototype.

The topmost added functionality resides in implementing a particular software developed in parallel with the theoretical research and screening for analysing and correlating the measurements. The developed prototype includes a more sophisticated method of analysis: the load distribution is measured not only as a difference between the right and left leg but was distinguished, for instance, through a limited number of pressure sensors, specifying the load on the different areas of the foot, the outside and inside edge load. Setpoints were established based on the existing standards for bodyweight and plantar pressures and the mean values of the data collected during the validation testing. The analytical algorithm (Table 1, Figure 3) enables the ENP system to calculate a segregated percentage for each load cell, self-calibrate before measurements, compare the recorded values with the established threshold, and display the results in a user-friendly manner for optimal evidence-based screening, early warning, and indication. Furthermore, the ENP can be used offline. This feature allows the use of the ENP by either patients or medical professionals in rural and remote settings with no access to the internet to measure their body balance for ad hoc evaluation. The cutoff point for all the cells was set to establish the screening’s colour codes. The mean values of the measurements by each pressure cell in the system were considered. A variation of 3% from the calculated average was further introduced for the upper and lower limits to set the warnings for above (red colour) and below (blue colour) levels, respectively. The colour codes are based on the average values obtained for each load cell upon tests performed three times for each subject for 50 subjects (e.g., the average for the right heel after 3 × 50 measurements is 9.8%).

This highly sensitive platform can screen for body imbalances possibly caused by the foot arch type as a home use device. Therefore, it can issue a timely warning message to the user and indicate the need for specific tests to rule out possible foot arch- and posture-related conditions.

### 2.3. The Foot Arch and Balance Study

Since the structures within the musculoskeletal system, bones, muscles, ligaments, and tendons, maintain the correct posture and respond to external factors such as footwear and loaded bags, monitoring these factors could trigger early awareness in young individuals before any related anatomical changes become permanent.

#### 2.3.1. Demographics

The parallel study regarding the foot arch and balance was conducted in a population of 1034 healthy young adults (18–23 years old) with no orthopaedical or podiatric medical history, including no injuries and no deviation of the foot arch.

#### 2.3.2. Data Acquisition

This study describes the frequency of foot arch types and metatarsal pressure on both right and left feet. It also investigated the relationships between a few anthropological parameters such as the bodyweight, height, pressure distribution on forefoot and rearfoot, and body posture in 54 subjects of the same age (20 years old). Three foot arch scans (IStep foot scanner; Aetrex technology) were performed for each subject enrolled to identify the foot arch types (high, medium, or low foot arch). A wall-mounted Posture Chart was used to analyse the subjects’ body postures before and while carrying schoolbags. Three measurements were performed for each subject and each school bag type used. The three school bag styles used were the most popular among students (backpack, tote bag, and sling bag). Figure 4 describes the protocol of the study. The pressure sensors within the iStep electronic platform measured the amount of pressure one exerted on the contact surface and identified the foot arch type (low, high, or medium), while the wall-mounted posture chart and the Posture Screen^TM^ mobile phone application provided the measurements for the posture evaluation. The anatomical landmarks observed to evaluate posture are presented in Figure 4. The study implied proper informed consent from the subjects and ethical approvals, which assured strict confidentiality of information and data.

#### 2.3.3. Data Analysis

Observational and inferential statistical analysis was used to evaluate the frequency of higher metatarsal pressures and the correlations between the shoe size and the type of foot arch, the loaded torso and the anatomical landmarks’ deviations, and the changes in the front and back plantar pressures.

## 3. Results

The ENP relies upon the Working Principle of the Weighing Scale and

compares the load distribution between the two feet of a user,calculates the relative imbalance of the load distribution,issues an early warning if measurements exceed certain thresholds (the range of acceptable norms derived from the research data),ensures transmission of data in an IoT manner.

### 3.1. The Prototype

Figure 5a presents the results of a simulation performed at a specific date and time transferred on a mobile phone and explains a colour-coded imbalanced weight distribution. The red and blue colours represent the warning given by the calculated imbalance. Even though the average is balanced, the detailed measurement describes the unevenness and raises awareness. Since the colour code is the most explicit reading (Figure 5a), the results on mobile applications allow for the rapid identification of the direction of the body sway towards the left.

The alternative display of weight and percentage charts is from the website. These diagrammatic representations in Figure 5b,c detail the distribution of the user’s weight at a specific date and time and explain the charts of weight and percentage. For instance, for (3) in Figure 5b, the software shows a weight distribution of 40 kg out of a total bodyweight of 77.7 kg. This point (3) is on the Toe section, and a similar reading method applies to the other sections (e.g., Heel). There are eight points of weight distribution showing a purple colour indicating a deviation towards the left: the Left Toe and the Left heel points and the overall deviation (Right and Left points, Toe and Heel points).

This chart is a per point display based on the total weight measured, and the accuracy is verified following the summations:Left + Right = Total Weight;(1)
Toes + Heel = Total Weight;(2)
L. Toes + L. Heel + R. Toes + R. Heel = Total Weight(3)

The formulas above describe the working principle of the ENP: it is a two-weight scale concept with eight-point load cells compared with the four points in a classic weight scale. The measurement result includes the values from the left and the right scales as in Formula (1). Similarly, Formula (2) shows that the total weight can be extrapolated from the loads measured by the cells at the Toes and Heels. Further analysis of the values corresponding to both sides of Toes and Heel (left and right) also leads to the total weight as presented by Formula (3). Figure 5c explains the website display in a percentage chart for the same subject, according to the sections Toes (forefoot) and Heel (rearfoot) and the right and left sides (Left, Right). The outer circle represents the sections corresponding to the Toes (forefoot) and Heel (rearfoot) and the sections depicting the right and left feet (Right, Left), while the inner circle shows the right toes, right heel, left toes, and left heel. For instance, (5) shows a shift towards the left side as its area is larger than (3), and (7) shows a shift on the left heel as it is larger than (4), the right heel.

The colour code is for each section corresponding to the right and left toes, heels, and right and left sides. For instance, if more than the average + 1.5% was recorded by a cell, the result was considered a warning for that pressure cell (Figure 5a: Left Heel). A value more than average + 3% for two cells on the same side (e.g., left) also triggers the awareness (Figure 5a: Left). Similarly, the blue areas indicate a warning if an average - 1.5% for one pressure cell and an average −3%) for two pressure cells on the same side are measured (Figure 5a: Right Heel and Right, respectively). The values were used to set the threshold for the system to be a warning indicator and help the user recognise the deviations and decide whether to check for early specialised advice. (Table 1) Repeated measurements using the ENP and the commercial scales and the calculated Standard Deviation (SD) for the scanners showed a 0.14 deviation for the ENP compared to the commercial weighing scale stated at 0.1 deviations (SD 0.10). We proved that our prototype is off by 0.03 compared to the commercial weight scales in this testing. We found that some scales significantly differed in weight output by about 1.3 kg+/− during the testing with multiple commercial weighing scales.

The relationship between the prototype and commercial equipment demonstrated via *t*-test indicated the measurement stability, with no significant differences in the measurements between the prototype and the commercial scanners (Appendix A: Appendix A).

The device uses the load cell to measure the weight distribution of the force applied. Bluetooth technology allows the data display on a mobile phone, and Wi-Fi technology ensures the data transfer to a backend database for further analysis and adaptation for multiple data monitoring. With a proper industry fabricated PCB, the final prototype achieved 0.1–1 kg deviation readings.

The results show that the ENP makes at-home monitoring of bodyweight and load distribution possible. Since the ENP can work as both a screening and disease monitoring tool, its use may vary from one user or patient to another. For instance, the user will follow the specialists’ instructions on how frequently to run the measurements based on the diagnostic and therapeutic schemes. Otherwise, if the ENP is employed as a prophylactic tool, users without any related medical conditions will measure their bodyweight and load distribution monthly.

### 3.2. The Foot Arch and Balance Study

The study conducted in parallel on foot arch and balance described the frequency of foot arch types and metatarsal pressures in a population of young adults (1034 healthy subjects). This study introduces data to support the need for early balance monitoring and supports the observation that daily tasks such as standing or carrying bags can influence posture. Therefore, in the long run, these effects could be related to improper posture and biomechanics-related locomotor diseases. Figure 6 presents the foot arch types in a young population and the metatarsal pressures. Furthermore, the study showed that smaller shoe size is related to the metatarsal pressure.

The bodyweight distributions on the right and left feet were measured and no significant difference between the right and left feet was observed in the studied population (t = 0.9 for 0.05 significance level).

Independent measurements presented the relationship between the type of school bag and the change in body posture in a population of 54 young, healthy subjects (18–23 years old) to strengthen the importance of screening and analysis of body posture in a population at the personal level, and to reinforce self-monitoring and healthy habits. Figure 7 depicts the linear regression models to indicate the relationship (regression analysis) between body height, weight, and pressure distribution on the forefoot and rearfoot while standing on the pressure mat.

The observations in Figure 7 show no strong correlations (weak correlation with R values close to zero) between the anthropological parameters (body height and weight) and posture (forefoot and rearfoot pressures). Figure 7a shows a weak negative correlation and explains an inverse relationship between the two associated parameters: the bodyweight increases while the forefoot pressure decreases. Therefore, a lower forefoot pressure tends to be associated with higher bodyweight. Figure 7b–e shows the weak positive correlations and the direct relationships between the variables observed. For instance, there is a direct association between the rearfoot pressure and bodyweight and height, respectively, as seen in Figure 7b–d. Furthermore, the slight body inclination towards the higher rearfoot weight-bearing percentile means that the subjects placed more weight on their rearfeet than on their forefeet.

Similarly, body posture was analysed by observing the load shift in the torso and pelvic balance on the posture chart. The measurements were conducted in correlation with school bag wearing of a backpack, tote bag, or sling/crossbody bag. The results indicated the positive correlations between the shift in the torso and pelvis while carrying the bags.

Similar to Figure 7 and Figure 8 shows direct relationships between the variables observed. However, the correlations are stronger. For instance, the change in torso posture is strongly associated with the change in the pelvis posture; therefore, there is a strong tendency, when carrying either a tote or sling bag, for a high degree of posture change in the torso associated with the high degree of posture change in the pelvis. The stronger the correlation, the stronger the relationship and deviation induced by carrying the bag which acts as a trigger.

## 4. Discussion

Generally, assessing body balance includes static, quasi-mobility, and mobility tasks. Since they often complement each other, numerous assessment methods have been developed in recent decades with complex and corresponding balancing devices and measurement tasks. However, most of them, which serve as diagnostic tools, are complex and require professional skills for interpretation [17,25,26,57,58]. Others may serve as balance training and rehabilitation with incorporated personalised tasks of varying difficulties [28,56,59,60,61]. The presented prototype was designed to help consumers self-measure their body balance in the same way a simple bodyweight scale helps. Several factors may influence the results when a scanner is used at home. For instance, the subjects’ ability to follow instructions during testing significantly affected the data collection. Therefore, a simple process to guide the user was created. The User manual comprises simple instructions for the subject: “(1) Step on the platform as shown in the user manual (the middle of your right and left foot over the crosslines and your toes at the drawn lines that show your feet size), and (2) Stand still and relaxed, arms parallel to the body while the scanner measures you.” The established guideline allowed the user to understand where to place the feet while standing on the platform. For instance, Figure 3b shows a line to guide and centre the subject as much as possible for the measurement. This step-by-step procedure was validated and the subjects quickly followed it, improving the accuracy of the measurements significantly. The user guide assists with a correct log in into the system for data privacy and protection: the user will login into the system with a username and password set previously for a virtual private network (VPN).

Furthermore, a clear display and easy access to data makes the scanner user-friendly. The ENP resembles a standard bathroom scale with additional functionality, mainly readings on the display window with Bluetooth or Wi-Fi data communication to a personal electronic device. The design of the ENP is straightforward to address patients’ immediate and easy measurement needs, compared with the complex and multifunctional analytical instrumentation available which is used in specialised clinics for professional investigation. The ENP distinguishes the distributed weight through a limited number of pressure sensors outside and inside edge load, repetitive tests, or tests with a determined position of the feet (e.g., standing position). Moreover, it specifies the fore- and rearfoot loads and displays them in various forms for easy reading and understanding of their significance. The colour codes establish fast recognition of the body balance status, so the user immediately ascertains any changes and decides to either store and monitor locally or transmit the data to a specialist.

Meanwhile, the map displayed shows the weight shifts and guides the user towards instant readjustment. For instance, the user will visualize the weight distribution on both sides, right and left or forward and backwards. This distribution directly indicates an imbalanced body and signals risks of falling and injuries [62,63,64]. Therefore, by indicating such changes, the developed prototype provides a home triage system for personal use, in combination with its standard weighing scale functions, to issue the recommendation to see a specialist and to monitor the impact of corrective measures (e.g., use of orthopaedical insoles/foot orthotics, specific fitness programs, or kinesiotherapy) [65,66]. Therefore, due to the main added functionality that resides in the analysis of the measurements, the scanner can be either a fast-screening tool (e.g., in the context of school health services for students), a simple therapeutic aid (e.g., to monitor the progress due to rehabilitation, physiotherapy, use of orthopaedical insoles, or other more complex chiropractic treatment), or both.

Another factor influencing healthcare is how data is transmitted and processed. The ENP prototype as an IoT-like device connects the user to healthcare providers for real-time monitoring and therapeutic decisions. The recorded data transmission is secured via protected channels. For instance, the users will activate a virtual private network (VPN) when they create a password-based login into the system prior to starting any measurement with the ENP. Furthermore, the medical systems are protected by special software to ensure the confidentiality of the data. Therefore, when using these channels, the medical specialists in the clinical settings will secure patient registration and medical information. This belongs to the larger group of remote healthcare assistance which support personalised diagnosis and treatment and specialised medical programs for remote areas or during at-home isolation imposed by epidemiological surveillance (e.g., the COVID-19 pandemic).

Since monitoring posture is one simple observational method, it is a task that, once performed at home, could teach healthy individuals or patients with biomechanical-related conditions about any changes in body balance and possible causes such as the foot arch type. The foot arch and posture study highlighted that 31% of subjects were high archers and 15% were flat footers in the young, healthy population. However, the results showed slight backwards leaning to maintain balance at the expense of body stability. Therefore, the posture trends (the regression model) in the observed population stress the need of knowing the type of foot arch. For instance, young and healthy people with a deviation from a neutral foot arch who are aware of the consequences of this foot type (e.g., tendonitis and plantar fasciitis) can improve the mechanics in the ankle with adequate footwear to maintain the correct posture and musculoskeletal system homeostasis [67]. Such examination opens new avenues for a clearer understanding of how daily tasks can influence the body posture immediately or over time. 

The regression models presented the correlation between the bodyweight and the two main plantar foot pressures, forefoot and rearfoot, under observation. When a correlation is measured, the two observed variables develop in ways that may explain the body’s responses to changes in the body balance and the tendency to adapt; the deviation from a balanced posture may be in the same or opposite direction. For instance, the negative correlation in Figure 7a, even though weak, shows no necessary tendency to readjust the weight distribution and load the forefoot pressure (kPa) when the bodyweight increases. This observation could be discussed in the context of the load distribution between the forefoot and rearfoot, concerning the bodyweight. The positive correlation, even though weak, in Figure 7b,e presents the tendency to load the bodyweight on the rearfoot, which could affect the body balance and thus walking. Such relationships conclude the importance of monitoring the bodyweight and posture for timely and adequate measures under medical supervision, as, over time, any small deviations within the locomotor system may lead to structural changes and impaired balance and gait. Postural and dynamic instability could increase the risk of falls, especially in an ageing population [68,69]. Falls are among the most common causes of injury, severe health problems, and even death in older adults. Numerous studies have revealed a relationship between falls and risk factors such as advanced age, declined cognitive function, strength deficit, gait abnormalities, and reduced balance [70,71,72,73]. Therefore, the causes of falls must be identified to predict their risk. Furthermore, it is necessary to quickly identify the leading factors causing falls through gait and posture tests and use the data to prevent falls. Novel methods are required to overcome the limitations of existing studies.

Furthermore, the relatively strong positive correlations between schoolbags and the plantar pressures in Figure 8 showing that the heavier the schoolbag is, the higher the plantar pressure is, present incipient information regarding the influence of wearing schoolbags on the body posture of young, healthy individuals. It has been observed that long-term incorrect posture may affect the morphological development of the spine with systemic severe consequences such as thoracic deformities and impaired lung function [74,75,76].

The observed tendency of leaning backwards while standing and the relationship between the metatarsal pressure and the type of foot arch [77,78,79] emphasize the importance of monitoring body posture to prevent pain and disability while ageing [80,81,82]. Therefore, the younger a person is, the higher the awareness should be through constant monitoring of the posture and metatarsal pressure, and the more positive the consequences on the general health status will be in respect to physical abilities while ageing [83,84].

Even though the significance of the correlation varies, the information transmitted contributes to a person’s knowledge and decisions regarding the maintenance of a healthy posture by carrying adequate types of bags correctly. Demonstrating the influence of carrying bags on body posture also contributes to better education of young individuals [85]. Medicine, and the concept of personalised care, are tailored to the individual patient’s needs based on their clinical background, disease prognosis, or assessed risks. Despite complex and multifactorial difficulties, considerable progress has been made in IoT and personalised medicine fields to improve and consistently manufacture reliable and stable systems that integrate sensing modalities [34,86,87]. The presented features could embed the developed platform into the emerging domain studies, aiming to capture human physiological parameters through wearable systems developed on flexible substrates (e-skin) [88]. Skin-like tactile sensing detects various stimuli such as pressure, strain, temperature, vibration, and sliding, and it can be used to monitor physical activity and position or to detect vital signs, such as blood pressure and respiratory rate. Other applications for monitoring coughs, abdominal breathing, elbow or finger [89] bending, walking, body motions, foot pressure, and body heat can be recorded via flexible sensors which are battery-free and interconnected to a wireless reader to permit remote health detection [90]. The unique features of e-skin contribute to the developing potential as an in situ diagnostic tool for further implementation in clinical practice at patient levels. Therefore, a robust interdisciplinary approach that combines micro-/nanoelectronics, material science, biotechnology, data transmission, and data processing technologies can develop alternatives to bulky diagnostic devices and is the next target in human health and prophylaxis [90,91]. In this direction, the developed ENP belongs to the systems that continuously investigate the risk factors and introduce IoT-based new devices to help monitor biomechanics-related conditions within populations at risk.

Therefore, cost-effective scanners for easy, effective at-home monitoring of bodyweight and weight distribution for the fast understanding of any slight imbalances are crucial for a timely clinical examination, diagnosis, and adequate intervention.

## 5. Conclusions

Determining the abnormal postural sway associated with an increased risk of falls requires access to laboratory equipment for diagnostics and monitoring. The newly created ENP as a proposed screening platform for the at-home measurement and monitoring of bodyweight and foot load distributions separated right from left, adds value to the current procedural and therapeutic aspects from the consumer perspective. The easy-to-use platform was designed with a self-check calibration error function that runs during its start-up to ensure that every measurement is accurate and provides reliable information to the users at home and to professionals who monitor the condition’s progress and amend the treatment. The developed platform also presents features that assess the user’s balance and bodyweight to permit a straightforward representation of the user’s weight distribution and load distribution on the right-left, front-back foot. Since the data collected is displayed as a simple numerical report and a map, the evaluation could be done at two levels for non-specialists and professionals. Moreover, the possibility of the real-time interpretation of the reported physiological parameters opens new avenues toward IoT-like on-body monitoring of human physiological signals through easy-to-use devices on flexible substrates for specific versatility. Clinicians also may exploit distance communication to monitor their patients’ body-worn postures via at-home, low-cost alternatives, miniaturised or not. Such solutions may be the next generation of user-friendly at-home devices designed for early diagnosis and personalised treatment and monitoring. Furthermore, posture-related data is the key to developing artificial intelligence in the field and personalised medicine. Therefore, the Early Notice Pointer is an IoT-like platform for point-of-care feet and body balance screening for early postural changes and fast data sharing that allows timely and accurate medical decisions, easy monitoring, and prophylaxis of locomotor system-related diseases.

## Figures and Tables

**Figure 1 micromachines-13-00682-f001:**
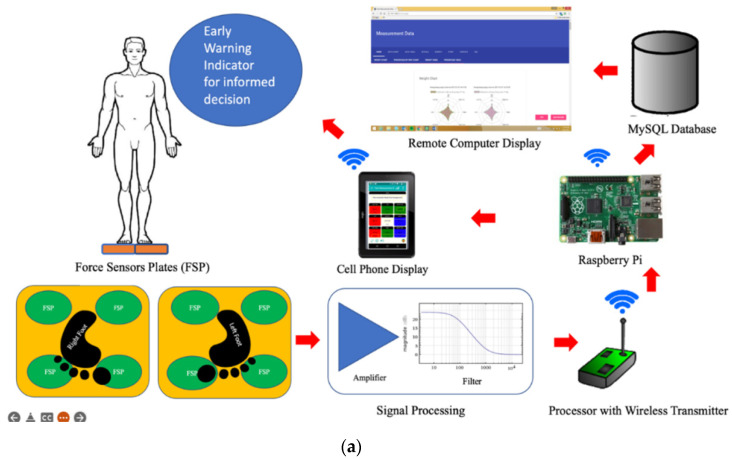
The design of the ENP: (**a**) an overview of the developed system, the segregation of eight different regions of the foot arches, and the placement of the sensory cells within the force sensor plates for signal processing, Wi-Fi transmission, iCloud databasing, and displaying and transmitting results; (**b**) the schematics of the electronics with one load cell connection exemplified.

**Figure 2 micromachines-13-00682-f002:**
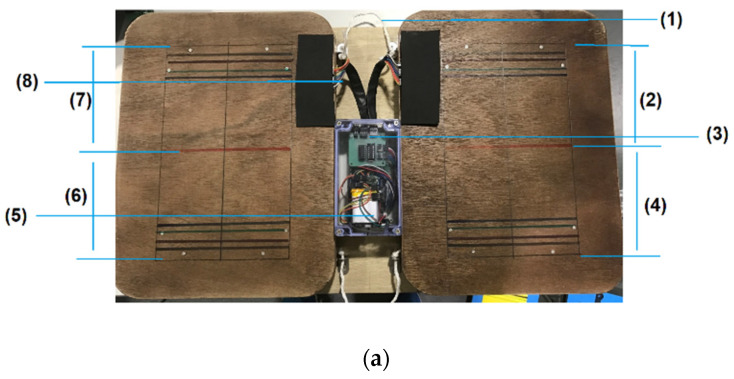
(**a**) The simple architecture prototype used for standing and measuring bodyweight load distribution on both feet: (1) inter-plate connection; (2) area of the platform that corresponds to the right forefoot that corresponds to pressure sensor cells; (3) electronics; (4) area of the platform that corresponds to the right rearfoot that corresponds to pressure sensor cells; (5) electronics; (6) area of the platform that corresponds to the left rearfoot that corresponds to pressure sensor cells; (7) area of the platform that corresponds to the left forefoot that corresponds to pressure sensor cells; (8) electronics; and (**b**) the testing process that starts with the user checking their feet size using the measuring template on the plate.

**Figure 3 micromachines-13-00682-f003:**
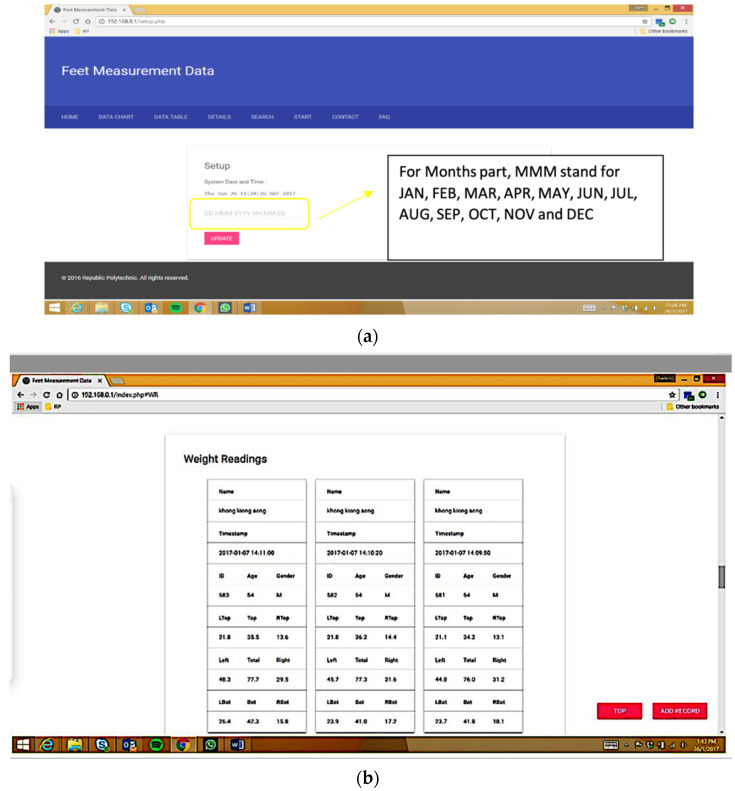
The website mode of ENP: (**a**) the Initiation and setup; (**b**) the Table page with multiple measurements to add.

**Figure 4 micromachines-13-00682-f004:**
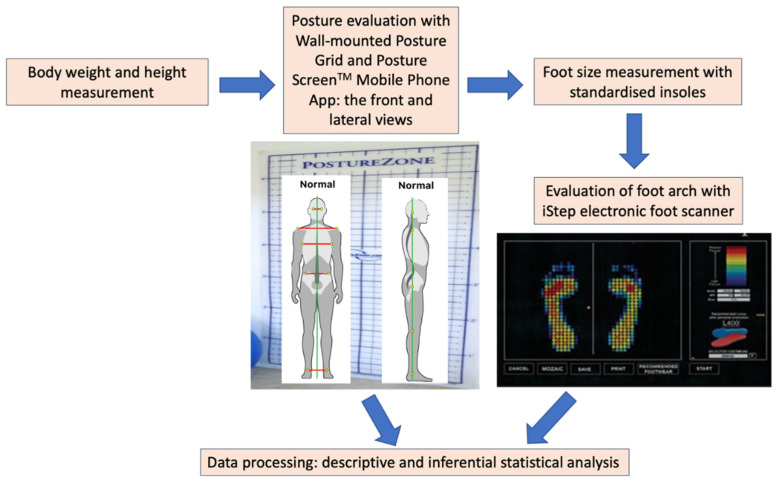
The investigation protocol for foot arch and posture evaluation.

**Figure 5 micromachines-13-00682-f005:**
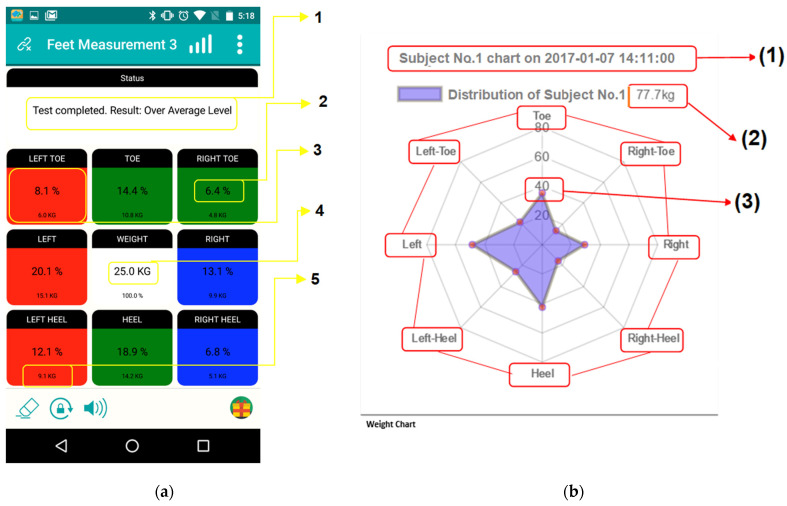
The software displays for (**a**) the mobile App-based display, indicating an unbalanced distribution of bodyweight loads right-left and front-back; (**b**) the website display in weight chart; and (**c**) the percentage chart for various sections (1) Right and (2) Right Toes, (5) Left and (6) Left Toes, (3) Right and (4) Right Heel, and (7) Left Heel and (8) Heel.

**Figure 6 micromachines-13-00682-f006:**
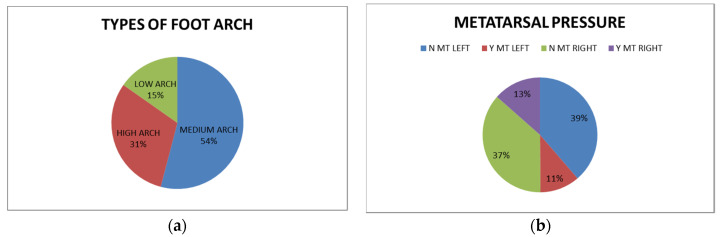
(**a**) The frequency of foot arch types; (**b**) The presence of metatarsal (MT) pressure in subjects in a young population.

**Figure 7 micromachines-13-00682-f007:**
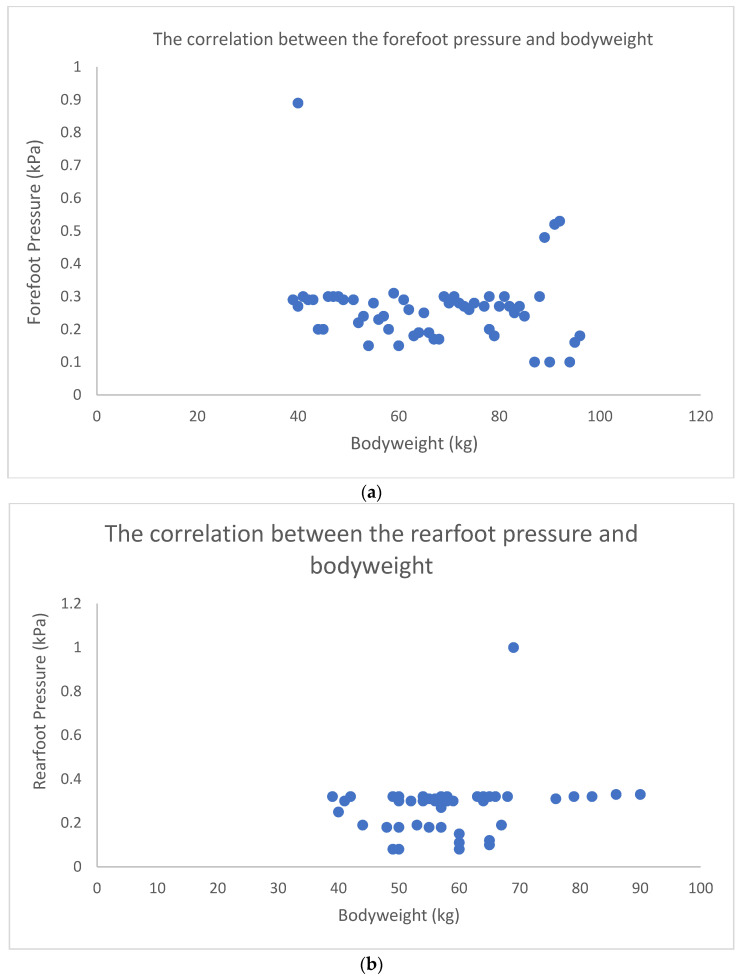
(**a**) The correlation of forefoot pressure (kPa) and weight (R = 0.13); (**b**) the correlation of rearfoot pressure (kPa) and weight (R = 0.16); (**c**) the correlation of forefoot pressure (kPa) and height (R = 0.16); (**d**) the correlation of rearfoot pressure (kPa) and height (R = 0.13); and (**e**) the tendency of weight bearing on rearfoot and forefoot (R = 0.12).

**Figure 8 micromachines-13-00682-f008:**
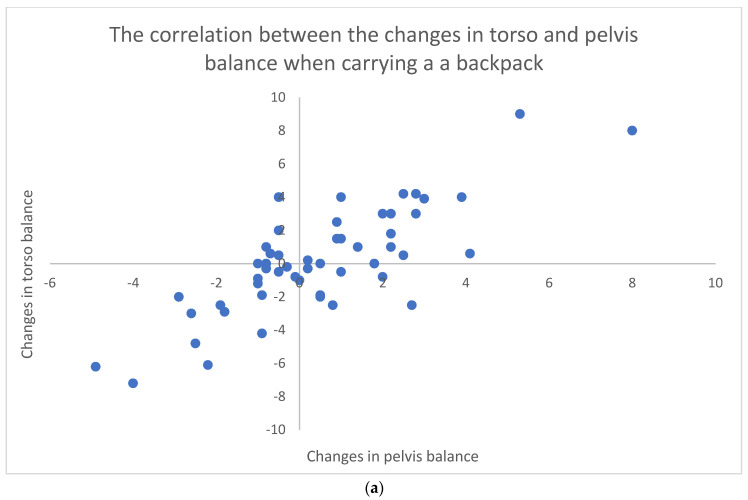
(**a**) The correlation between torso and pelvis when carrying a backpack (R = 0.82); (**b**) the correlation between torso and pelvis when carrying a tote bag (R = 0.49); and (**c**) the correlation between torso and pelvis when carrying a sling/crossbody bag (R = 0.94).

**Table 1 micromachines-13-00682-t001:** The colour codes are based on the threshold for bodyweight percentage distribution detected by the load cells inside the platform.

Red Point >= 2: Above Average Level		
Blue Point > 2: Below Average Level		
Red Point < 2 or Blue <= 2: Average Level		
>=7.9%	>=16.1%	>=7.9%
<=4.9%	<=10.1%	<=4.9%
Best = 6.4%, <7.9% And >4.9%	Best = 13.1%, <16.1% and >10.1%	Best = 6.4%, Not < 7.9% and >4.9%
Any correct colour value will be given 1 point	Any correct colour value will be given 2 points	Any correct colour value will be given 1 point
>=19.8%	Weight Value	>=19.5%
<=13.8%		<=13.5%
Best = 16.8%, <19.8% And >13.80%		Best = 16.5%, <19.5% and >13.5%
Any correct colour value will be given 2 points		Any correct colour value will be given 2 points
>=11.6%	>=23.2%	>=11.3%
<=8.6%	<=17.2%	<=8.3%
Best = 10.1%, <11.6 And >8.6%	Best = 20.2%, <23.2% and > 17.2%	Best = 9.8%, <11.3% and >8.3%
Any correct colour value will be given 1 point	Any correct colour value will be given 2 points	Any correct colour value will be given 1 point

## Data Availability

The raw and processed data required to reproduce these findings cannot be shared at this time due to technical or time limitations. The raw and processed data will be provided upon reasonable request to anyone interested anytime after the technical problems are solved.

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
