# Peer review of "Early Notice Pointer, an IoT-like Platform for Point-of-Care Feet and Body Balance Screening"

_micromachines, 2022, doi:10.3390/mi13050682_

Round 1

Reviewer 1 Report

Paper ID # micromachines-1658284

The topic looks interesting to me. However, I have a few comments to improve the quality of the manuscript. My comments are as follows:

Comments:

  1. The article has minor grammatical and punctuational issues that need to be addressed.
  2. The authors add twice “The design of the prototype” subsection under sections 2 (Materials and Methods) and 3 (Results). It is suggested to remove this subsection from section 3 and combine it with the subsection of section 2. Results should be focused on only the results and their discussion, not the measurement protocols or the system designing approaches.
  3. The authors applied the student t-test for validating their products against two commercial foot scanners (Matscan and RS FootScan).         a) What is the sample size of this validation test?                                  b) If it is below 30, the student t-test may not be useful and preferred. In that case, chi-squared or other tests could be preferable.
  4. While I started reading the introduction and materials and method sections, it looked interesting to me. However, I feel frustrated to see the results. I didn’t find any elaborate explanation of their results (e.g., Figures 7 and 8).
  5. In Figure 7, the author claimed that there was no correlation between the anthropological parameters (body height and weight) and the posture (forefoot and rearfoot pressures). It is not clear what is the significance of adding these plots to the manuscript.
  6. Also, the significance of this article was vaguely stated in the conclusion.

Author Response

Thank you for your time and comments, please find the answers as it follows:

  1. The article has minor grammatical and punctuational issues that need to be addressed.

Response:

We checked the editing and grammar and modified accordingly.

  1. The authors add twice “The design of the prototype” subsection under sections 2 (Materials and Methods) and 3 (Results). It is suggested to remove this subsection from section 3 and combine it with the subsection of section 2. Results should be focused on only the results and their discussion, not the measurement protocols or the system designing approaches.

Response:

We modified accordingly: from line 358-363 to Line 216-221

The proposed sensitive and easy to use screening platform has been prototyped (Figure 2) and tested for consistent data acquisition. A mobile application (App) and a website display were developed to present the testing results. Figure 3 illustrates the mobile App-based and website-based displays. The colour code display is the most advanced and allows the results recorded by the device to be easily interpreted by the user and the specialists.

  1. The authors applied the student t-test for validating their products against two commercial foot scanners (Matscan and RS FootScan).         a) What is the sample size of this validation test?                                  b) If it is below 30, the student t-test may not be useful and preferred. In that case, chi-squared or other tests could be preferable.

Response:

As highlighted in 2.2 The testing of the prototype/Demographics: Line 251-253.

The cutoff was set to determine the level of distribution based on the data collected from 150 subjects (normal distribution in a healthy population, the sample size is sufficient for the statistical significance of the pilot study).

  1. While I started reading the introduction and materials and method sections, it looked interesting to me. However, I feel frustrated to see the results. I didn’t find any elaborate explanation of their results (e.g., Figures 7 and 8).

Response:

We added further explanations to Results and Discussions with reference to Figure 7 and 8 and the concept of statistical correlations to answer the Comments 4 and 5:

Line 475- 485.

The observations in Figure 7 show no strong correlations (weak correlation with R values close to zero) between the anthropological parameters (body height and weight) and the posture (forefoot and rearfoot pressures). Figure 7(a) shows a weak negative correlation and explains an inverse relationship between the two associated parameters: the bodyweight increases while the forefoot pressure decreases. Therefore, the lower forefoot pressure tends to be associated with higher bodyweight. Figure 7 (b), (c), (d) and (e) show the weak positive correlations and the direct relationships between the variables observed. For instance, there is a direct association between the rear foot pressure and body weight, and height, respectively, in Figure 7 (b), (c) and (d). Furthermore, the slight body inclination towards the higher rear foot weight-bearing percentile, means that the subjects placed more weight on their rear feet than on their forefeet.

Line 498-504

Similar to Figure 7, Figure 8 shows direct relationships between the variables observed. However, the correlations are stronger. For instance, the change in torso posture is strongly associated with the change in the pelvis posture; therefore, there is a strong tendency: when carrying a bag, either tote or sling bag, the high degree of posture change in the torso tends to be associated with the high degree of posture change in pelvis. The stronger the correlation the stronger the relationship and the deviation induced by carrying the bag which acts as a trigger.

Line 578-596

Such examination opens new avenues of clearer understanding of how daily tasks can influence the body posture either immediately or in time. The regression models presented the correlation between the bodyweight and the two main plantar foot pressures, fore foot and rear foot, under observation. When a correlation is measured the two observed variables develop in ways that may explain the body responses to changes in the body balance and the tendency to adapt: the deviation from a balanced posture may be in the same or opposite direction. For instance, the negative correlation in Figure 7(a), even though weak, shows that there is no necessary tendency to readjust the weight distribution and load the fore foot pressure (kPa) when the bodyweight increases. This observation could be discussed in the context of the load distribution between the fore foot and rear foot and concerning the bodyweight. The positive correlation, even though weak, in Figure 7 (b) and (e) presents the tendency to load the bodyweigh on the rear foot, which could affect the body balance and thus walking. Such relationships conclude the importance of monitoring the bodyweight and posture for timely and adequate measures under medical supervision, as overtime any small deviations within the locomotor system may lead to structural changes and impaired balance and gait. Postural and dynamic instability could increase the risk of falls, especially in an ageing population. (Drew, T., Prentice, S. & Schepens, B. Cortical and brainstem control of locomotion. Prog. Brain Res. 143, 251–261 (2004); Aboutorabi, A., Arazpour, M., Bahramizadeh, M., Hutchins, S. W. & Fadayevatan, R. The effect of aging on gait parameters in able-bodied older subjects: A literature review. Aging Clin. Exp. Res. 28, 393–405 (2016). Falls are among the most common causes of injury, severe health problems, and even death in older adults. Numerous studies have revealed a relationship between falls and risk factors such as advanced age, declined cognitive function, strength deficit, gait abnormalities, and reduced balance. (Mancini, M. et al. Continuous monitoring of turning mobility and its association to falls and cognitive function: A pilot study. J. Gerontol. A 71, 1102–1108 (2016). 2. Fuller, G. F. Falls in the elderly. Am. Fam. Physician 61, 2159 (2000). 3. Muir, S. W., Gopaul, K. & Montero Odasso, M. M. Te role of cognitive impairment in fall risk among older adults: A systematic review and meta-analysis. Age Ageing. 41, 299–308 (2012). Robinovitch, S. N. et al. Video capture of the circumstances of falls in elderly people residing in long-term care: An observational study. Lancet 381, 47–54 (2013). [70-73] Therefore, the causes of falls must be identified to predict their risk. Furthermore, it is necessary to identify timely  the leading predictor factors causing falls through a gait and posture tests and use the data to prevent falls. Novel methods are required to overcome the limitations of existing studies. 

Line 603-608

Furthermore, the relatively strong positive correlations between the schoolbags and the plantar pressures in Figure 8, the heavier the schoolbag is, the higher the plantar pressure is, present incipient information regarding the influence of wearing school bags on the body posture in young, healthy individuals. It has been observed that long-term incorrect posture may affect the morphological development of the spine with  systemic severe consequences such as thoracic deformities and impaired lung function. (Aggarwal N, Anand T, Kishore J, Ingle GK. Low back pain and associated risk factors among undergraduate students of a medical college in Delhi. Educ Health (Abingdon). 2013;26(2):103–8); Lowe TG, Line BG. Evidence based medicine: analysis of Scheuermann kyphosis. Spine (Phila Pa 1976). 2007;32(19 Suppl):S115–9.) (SzczygieÅ‚ E, Zielonka K, MÄ™tel S, Golec J. Musculo-skeletal and pulmonary effects of sitting position - a systematic review. Ann Agric Environ Med. 2017;24(1):8–12.)

Line 618-621

Demonstrating the influence of carrying the bags on body posture also contributes to better education of young individuals. Wen, L., Lin, X., Li, C., Zhao, Y., Yu, Z., & Han, X. (2022). Sagittal imbalance of the spine is associated with poor sitting posture among primary and secondary school students in China: a cross-sectional study. BMC Musculoskeletal Disorders23(1), 1-12.

  1. In Figure 7, the author claimed that there was no correlation between the anthropological parameters (body height and weight) and the posture (forefoot and rearfoot pressures). It is not clear what is the significance of adding these plots to the manuscript.

Response: We added in the new version of the manuscript:

Line 446-449

This study introduces data to support the need of early balance monitoring. It supports the observation that daily tasks such standing or carrying bags can influence posture. Therefore, in the long run, the effect could lead to improper posture and biomechanics-related locomotor diseases.

Please find the answers to Comments 4 and 5 highlighted in Results and Discussions to explain that a weak correlation means a present correlation still. In statistics, a weak negative correlation is a weak negative relationship or a weak inverse relationship between the two variables explaining that higher values of one variable tends to be associated with lower values of the other, however the association is weak. Similarly, a weak positive correlation is still a positive relationship or a direct relationship between the two variables explaining that higher values of one variable tends to be associated with higher values of the other, however the association is weak. 

  1. Also, the significance of this article was vaguely stated in the conclusion.

Response: We further added for stronger conclusion.

Line 665-670

Furthermore, posture-related data is the key for developing artificial intelligence in the field and personalised medicine. Therefore, the Early Notice Pointer is an IoT-like platform for point-of-care feet and body balance screening for early detection of postural changes, fast data sharing that allows timely and accurate medical decisions, easy monitoring and prophylaxis of locomotor system-related diseases.

Reviewer 2 Report

Micromachines

1658284

Early Notice Pointer, an IoT-like platform for point-of-care feet 2and body balance screening

The manuscript "Early Notice Pointer, an IoT-like platform for point-of-care feet and body balance screening" describes how a newly developed ENP serves as a potential screening platform for at-home measurement and monitoring of body weight and how separated right from left foot load distributions add value to the current procedural and therapeutic aspects from the consumer’s perspective. The manuscript is well-written and structured, and the results shown are promising. Therefore, it might attract a wide readership or the audience's interest in Micromachines. Below are some comments and suggestions for improving the quality of the manuscript before its consideration for Micromachines.

[1] Do we need to calibrate the ENP each time before use? How to justify the ENP providing accurate and reliable results if the machine does not require calibration?

[2] Can we use the ENP without an internet connection? Many rural areas in developing countries do not have internet access.

[3] How do we ensure the security of personal data against leakage? Please explain in the main text.

[4] Please also mention in the main text how often a person should use the ENP prototype to look for signs of imbalance so that medical treatment can start as soon as possible.

Author Response

Thank you for your time and comments, please find the following details in the manuscript:

[1] Do we need to calibrate the ENP each time before use? How to justify the ENP providing accurate and reliable results if the machine does not require calibration?

Response: please find the following information

Line 303 - 307

The analytical algorithm (Table 1, Figure 3, 5) enables the ENP system to calculate segregated % for each load cell, self-calibrate before measurements, compare the recorded values with the established threshold and display the results in one user-friendly manner for optimal evidence-based screening and an early warning and indication. 

[2] Can we use the ENP without an internet connection? Many rural areas in developing countries do not have internet access.

Response: we added the following to the manuscript

Line 307 -309

Furthermore, the ENP can be used offline. This feature allows the use of the ENP by either patients or medical professionals in rural and remote settings with no access to the internet to measure the body balance for ad hoc evaluation. 

[3] How do we ensure the security of personal data against leakage? Please explain in the main text.

Response: we added the following to the manuscript

Line 524-527

The user guide will assist with a correct log in into the system for data privacy and protection: the user will login into the system with a username and password set previously for a virtual private network (VPN).

Line 556 -562

The recorded data transmission is secured via protected channels. For instance, the users will activate a virtual private network (VPN) when they will password-based login into the system prior to starting any measurement with the ENP. Furthermore, the medical systems are protected by special software to ensure the confidentiality of the data. Therefore, when using these channels, the medical specialists in the clinical settings will secure patient registration and medical information.

[4] Please also mention in the main text how often a person should use the ENP prototype to look for signs of imbalance so that medical treatment can start as soon as possible.

Response: we added the following to the manuscript

Line 435 -441

The results show that the ENP makes at-home monitoring of body weight and load distribution possible. Since the ENP can work as a screening and disease monitoring tool, its use may vary from one user or patient to another. For instance, the user will follow the specialists’ instructions on how frequently to run the measurements based on the diagnostic and therapeutic schemes. Otherwise, if the ENP is employed as a prophylactic tool, the users without any related medical conditions will measure the body weight and load distribution monthly.